# Photonics-Assisted Terahertz-Wave Beam Steering and Its Application in Secured Wireless Communication

Kazutoshi Kato 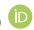

Graduate School of Information Science and Electrical Engineering, Kyushu University, 744 Motooka, Nishi-ku, Fukuoka 819-0395, Japan; kato@ed.kyushu-u.ac.jp

**Abstract:** Beam forming and beam steering are inevitable technologies for the practical application of high-frequency electromagnetic waves. Specifically, beam control technology using a phased array for terahertz waves above 100 GHz is necessary to realize the future of high-speed wireless communication. By photomixing, which is a promising method for generating terahertz waves, the phase of the generated waves can be tuned in the optical domain, so that the beam from the phased array can be controlled by photonics technologies. Directing the beam of a terahertz wave enables wireless communication to be improved not only via an increase in power efficiency but also in security in the physical layer of the wireless transmission. By utilizing this advantage and using coherent detection at the receiver, a secured wireless communication system is proposed, and the fundamental mechanism is demonstrated in a feasibility experiment.

**Keywords:** photomixing; photodiode; terahertz wave; beam steering; phased array; wireless communication

## 1. Introduction

Fifth generation (5G) technology using the 30-GHz band has recently seen global application, but already 6G/Beyond 5G technology, which utilizes much higher frequencies, is being researched and is attracting attention [1,2]. For this emerging technology, a frequency range of 100 GHz or higher is required, and terahertz waves are a candidate as a carrier. It is difficult for an electromagnetic wave with a higher power to be generated using current devices, and the attenuation of such an electromagnetic wave in the atmosphere is greater for a higher frequency. Therefore, the main challenge facing the implementation of wireless communication using terahertz waves, especially above 100 GHz, is attaining efficient transmission power. Beam forming (forming a beam into a specific direction) and beam steering (dynamically changing the beam direction) are inevitable technologies for meeting this requirement, capable of radiating electromagnetic waves and controlling their phases from arrayed antennas and combining them to form a beam in space [3–6].

Furthermore, photomixing, one of the generation methods of electromagnetic waves, generates high-frequency current as a beat signal with a frequency difference between two lightwaves by using the square-law detection characteristic of a photodiode [7–9]. In photomixing, the phase of the generated high-frequency current can be controlled by adjusting the phase of the lightwaves in the optical domain. This suggests the possibility to form and control electromagnetic waves using photonics technology.

By directing the beam of electromagnetic waves, the wireless communication becomes power efficient and secured in the physical layer, and by taking advantage of this characteristic and using coherent detection at the receiver, it is possible to enhance the security of wireless communication.

In this paper, we describe the technologies of arrayed photomixing and phase management, which are necessary to enhance the power and to control the beam of terahertz waves. In addition, as an application of the controlled terahertz-wave beam, we show a concept of a secured wireless communication system using coherent detection to make a heterodyned signal between two data sequences on the beams and demonstrate a feasibility experiment.

## 2. Beam Forming and Beam Steering Technology

For beam forming and steering electromagnetic waves, a phased array is generally used, in which the same signal is emitted from multiple antennas aligned in an array. Assuming that an array of $N$ element antennas is arranged on the $x$-axis at equal intervals $d$ as shown in Figure 1 and that each element antenna is activated with an equal amplitude of 1 and phase difference of an equal interval of $\Delta\phi$, the electric field amplitude of radiated electromagnetic wave $E_{beam}$ is expressed as the sum of the radiation intensity of each element antenna;

$$E_{beam} = \sum_{n=0}^{N-1} [\exp\{in(\Delta\phi - k_0 d \sin\theta)\}]$$
$$= \sum_{n=0}^{N-1} \left[ \frac{\sin\frac{Nu}{2}}{\sin\frac{u}{2}} \right] \qquad \left( d \leq \frac{\pi}{k_0} \right) \tag{1}$$

$$u = \Delta\phi - k_0 d \sin\theta \tag{2}$$

where $k_0$ and $\theta$ represent the wave number in free space and the angle along the $x$-axis, respectively. In a direction of $\theta = \sin^{-1}(\Delta\phi/k_0 d)$ or $u = 0$, $E_{beam}$ has the maximum value of $N$. Since the power is proportional to the square of the electric field, the peak power of the beam becomes $N^2$-fold. For beam steering, $\Delta\phi$ is changed so that $E_{beam}$ has the maximum value in a desired angle [10]. Although electric phase shifting is more reliable and faster than doing so mechanically, a more advanced circuit technology is required to control the phase at a higher frequency.

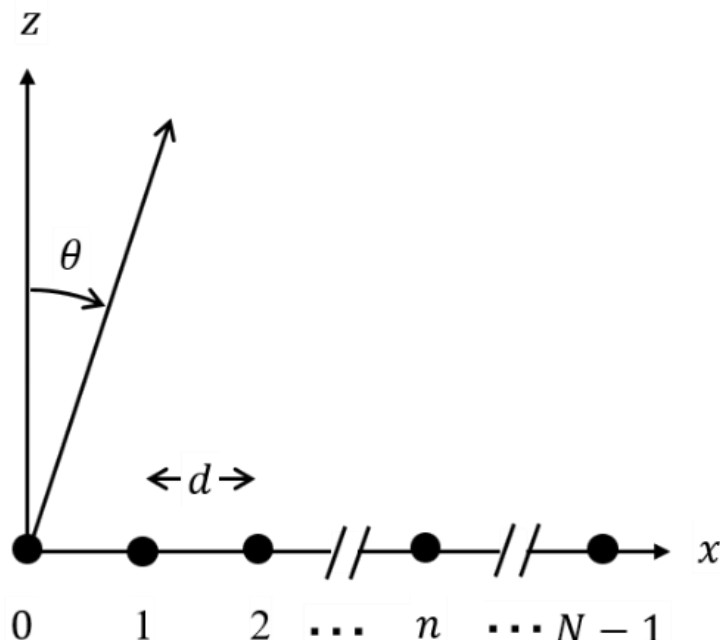

**Figure 1.** Layout of $N$ element antenna array.

In photonics technology developed for optical fiber communications, the phase of lightwaves can be controlled by using the refractive index change at the optical waveguide (electro-optic effect, thermo-optical effect, and carrier plasma effect). Fortunately, because photomixing is based on lightwave-to-electrical wave conversion, it is expected that the phase of a terahertz wave can be controlled at the optical domain. Thus, photomixing is considered to be an effective technology for controlling the terahertz-wave beam.

## 3. Photomixing and Phase Control

Photomixing can be used to convert two lightwaves of different frequencies into an alternating current with the same frequency as that of the difference of the two lightwaves at a photodiode (PD). This has been intensively studied and is considered to be a promising method for generating terahertz waves for wireless communications, as well as sensing and imaging technologies. The uni-traveling carrier PD (UTC-PD), which was developed originally as an ultrafast PD for optical communications, has higher output power saturation than that of conventional pin-PDs. However, even at the UTC-PD, the output power is saturated above a certain value because of the space charge effect at the carrier transport region. Arrayed UTC-PDs [11] are a solution for overcoming this issue, and therefore phase control at the UTC-PD is the next issue requiring attention.

As shown in Figure 2, at a photomixing system, two lasers with different frequencies are combined and photoelectrically converted at the PD to generate an alternating current with the same frequency as that of the difference between the two lightwaves. When the electric fields of the two lightwaves are $E_1$ and $E_2$, the power $P$ of the lightwaves detected by the PD is proportional to the square of the total electric fields of the two lightwaves. Thus, they are expressed as

$$E_1 = A_1 \exp\{i(2\pi f_1 t - k_1 x_1 + \varphi_1)\} \tag{3}$$

$$E_2 = A_2 \exp\{i(2\pi f_2 t - k_2 x_2 + \varphi_2)\} \tag{4}$$

$$
\begin{aligned}
P &\propto |E_1 + E_2|^2 \\
&= A_1{}^2 + A_2{}^2 + 2A_1 A_2 \cos\{2\pi(f_2 - f_1)t - (k_2 x_2 - k_1 x_1) + (\varphi_2 - \varphi_1)\}
\end{aligned} \tag{5}
$$

where $A_1$ and $A_2$ are amplitudes of the electric field of lightwaves, $f_1$ and $f_2$ are the frequencies, $k_1$ and $k_2$ are the wavenumbers, $\varphi_1$ and $\varphi_2$ are the phases of the light waves, and $x_1$ and $x_2$ are the optical path lengths. As a result of the square-law detection at the PD (lightwaves are detected as energy), an alternating current with frequency $f_2 - f_1$ and phase $-(k_2 x_2 - k_1 x_1) + (\varphi_2 - \varphi_1)$ is generated.

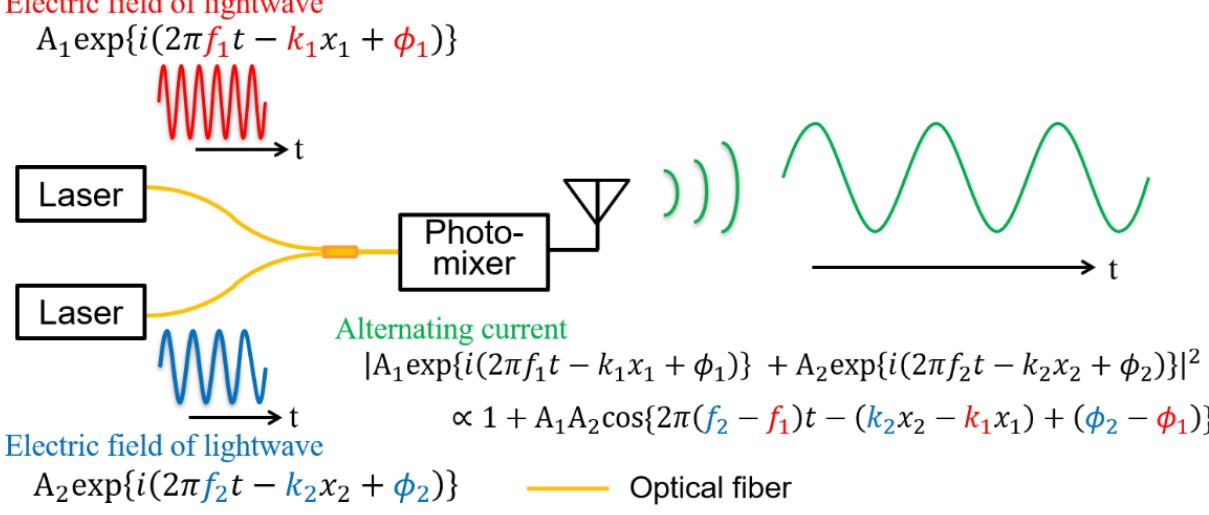

**Figure 2.** Photomixing system.

As described above, arrayed PDs which generate terahertz waves with the same phase can be combined to make a higher-power beam. To tune the phase of the terahertz waves, either $-(k_2 x_2 - k_1 x_1)$ or $(\varphi_2 - \varphi_1)$ should be changed. The former corresponds to changing the optical path length when the two lightwaves are in the same optical path, while the latter corresponds to changing the phase of each lightwave when the two lightwaves are in different optical paths.

In the former case, when both the path lengths are changed by $\Delta x$, which corresponds to a "true time delay", the change of the phase difference between the two lightwaves is given by the following equation.

$$- (k_2 - k_1)\Delta x = 2\pi \left( \frac{1}{\lambda_1} - \frac{1}{\lambda_2} \right) \Delta x \tag{6}$$

where $\lambda_1$ and $\lambda_2$ are the wavelengths of the two lightwaves. In the air, for two lightwaves in the 1550-nm band with an optical frequency difference of 300 GHz (wavelength difference of 2.4 nm), the value in parentheses is about 1 mm$^{-1}$. If an optical delay line (ODL) is used which changes the air gap between two collimator lenses, for example, to change both $x_1$ and $x_2$ by 1 mm ($\Delta x = 1$ mm), the phase of a terahertz wave is changed by $2\pi$, or one wavelength of a 300-GHz wave.

In the latter case, the same amount of phase change as that of the lightwave can be applied to the terahertz wave. For example, if a phase change of $\pi$ is applied to the lightwave, the generated terahertz wave will also have a phase change of $\pi$. Since an optical phase modulator can change the phase of a lightwave by about $\pi$ within an operating voltage, it is easy for the phase of terahertz waves to be changed by $\pi$. However, the latter method requires the use of optical components with optical fibers, such as optical phase modulators, and therefore phase stabilization technology is also required to suppress phase fluctuations in the optical fiber. This means that, for the latter case, a phase control device without optical fibers is required.

## 4. Terahertz Wave Beam Forming and Steering

To testify the photonic controllability of the terahertz wave, we made an arrayed photomixer with integrated terahertz antennas. As shown in Figure 3, four UTC-PDs are arranged in a direction parallel to the array with an interval of 0.5 mm, which is a half-wavelength of 300 GHz [12]. Each UTC-PD is connected to four slot antennas aligned in the direction perpendicular to the array so that a total of 16 antennas are arranged in a $4 \times 4$ two-dimensional configuration.

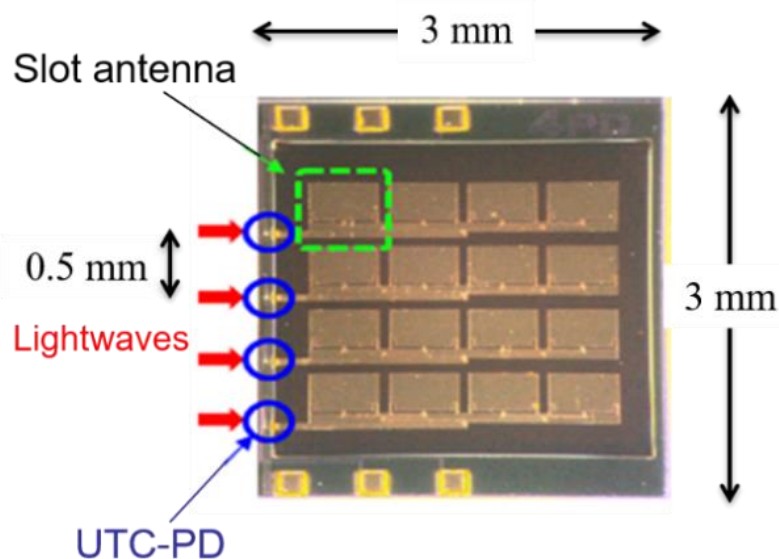

**Figure 3.** Arrayed UTC-PDs with integrated terahertz antennas.

First, phases were tuned by changing the optical paths with the ODLs. As shown in Figure 4, two lightwaves from lasers with an optical frequency difference of 300 GHz were combined by an optical coupler and then divided into four optical paths by an optical splitter. Each pair of lightwaves was introduced to each ODL and then to the UTC-PD aligned in the array. The intensity of the combined terahertz wave was measured

by the Schottky barrier diode (SBD) located at a distance of 100 mm from the chip [13]. The experimental setup is shown in Figure 5. Each ODL was tuned so that the detected intensity was maximized. Then, the SBD was rotated around the photomixers while maintaining the distance of 100 mm to obtain the angular distribution of the intensity of the combined terahertz wave. Figure 6 shows the measured intensity when one, two, and four photomixers were operated. The intensity increased as the number of operating photomixers increased, and a peak intensity of about 60 µW was obtained with four photomixers. It was confirmed that the peak intensity is proportional to the square of the number of elements, as expected. Next, using this configuration, each phase was tuned to steer the beam. For observing a beam steering, the SBD was set at a certain initial angle and the ODLs were tuned to maximize the detected power. Once optimized phases were found and fixed, the SBD was rotated to obtain the angular distribution of the intensity. Such measurements were made several times with different initial angles of the SBD [14]. Figure 7 shows the angular distribution of the power at each measurement. The peak was shifted by changing the phases, although the peak intensity decreases as the angle increases due to the directivity of the antenna itself. It was also found from these results that beam steering extends the 3-dB-rolloff angle up to 30°. The roll-offs and dips in the power distribution result mainly from the radiation pattern of antenna itself because the overall radiation pattern consists of the product of array factor and the radiation pattern of an antenna itself.

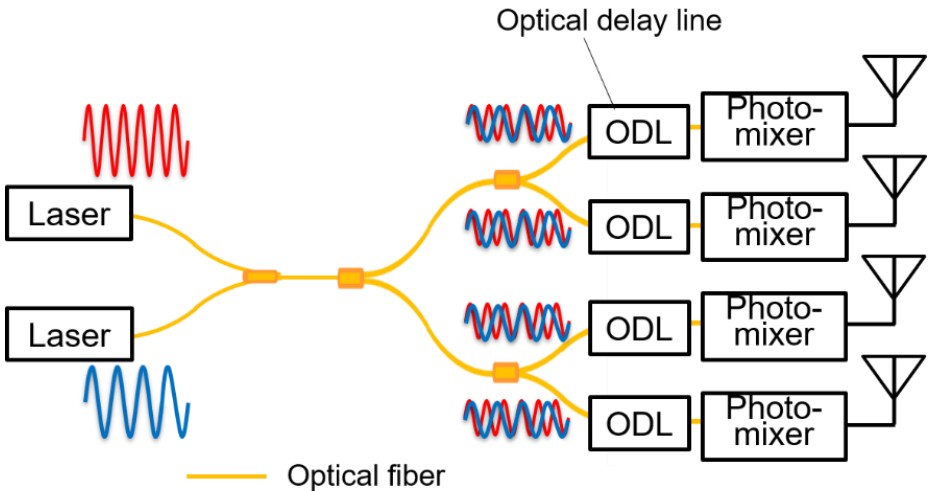

**Figure 4.** Configuration of 4-arrayed photomixing system.

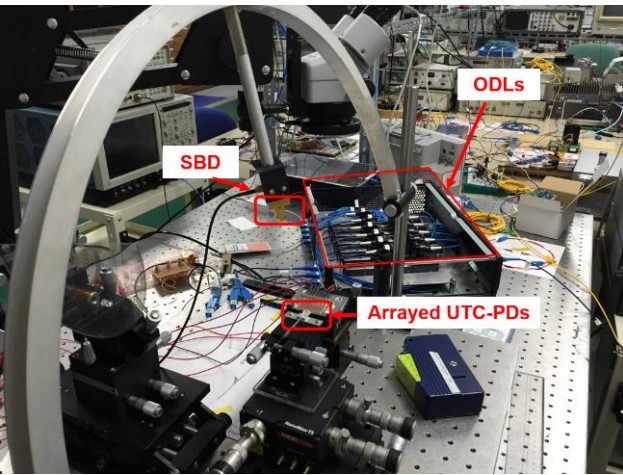

**Figure 5.** Experimental setup for measuring the power of terahertz wave.

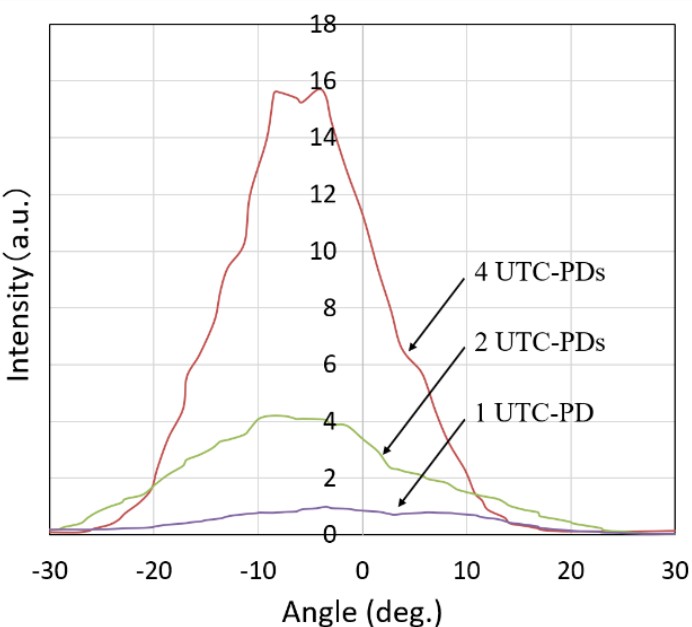

**Figure 6.** Measured intensity of 300−GHz wave.

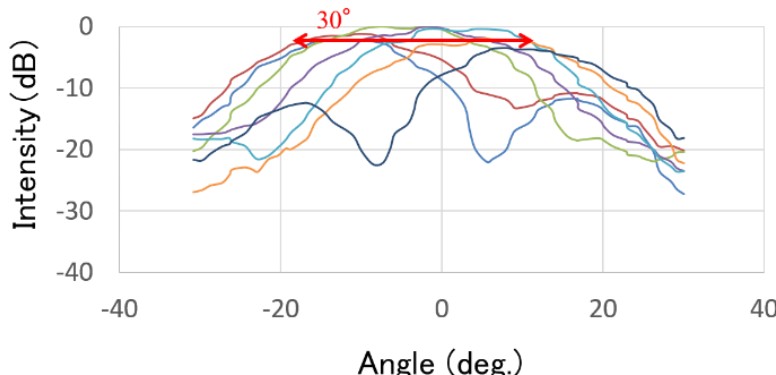

**Figure 7.** Measured intensity when the beam is steered.

Since phase tuning using the ODL changes the optical path length with the order of millimeters, it is a tolerant method to the fluctuation of optical fiber length. However, the ODL is a mechanically controlled component consisting of bulky parts and the control takes the order of seconds. The term $(\varphi_2-\varphi_1)$ in the phase component shown in (5) indicates that the phase change can be obtained by changing the phase of one of the two lightwaves traveling in different optical paths. To achieve this configuration, we developed an integrated phase controller consisting of a silica-based planar lightwave circuit as shown in Figure 8. One of the lightwaves inputted into this phase controller was split into eight optical paths, each of which had a thermo-optical phase shifter. Then, by using an integrated optical coupler, each lightwave was merged with the other lightwave, which was also split into eight optical paths. Since there were no optical-fiber-based components between the splitters and couplers, fluctuation of the optical path length was suppressed, which enabled stable phase tuning of the lightwaves. The thermo-optical phase shifter has a micro heater whose temperature is controlled by applying voltage, and the response time is determined by the thermal conductivity which is on the order of milliseconds [15].

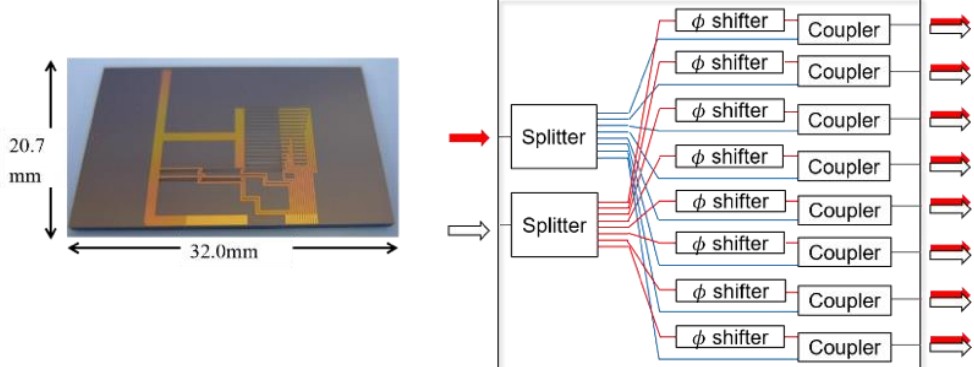

**Figure 8.** Integrated phase controller consisting of a silica-based planar lightwave circuit; photograph (**left**) and schematic configuration (**right**).

To testify the electronic controllability, stability, and fast response of the integrated phase controller, we operated it to steer the beam with a high repetition. Two lightwaves of the 1550-nm wavelength band with an optical frequency difference of 300 GHz were inputted to an integrated phase controller, and the two outputs were injected into two UTC-PDs in the arrayed UTC-PDs. A 1-kHz sinusoidal voltage (Figure 9, upper curve) with an amplitude of 18 V (corresponding to a phase change of $0.9\pi$) was applied to one of the channels of the integrated phase controller to dynamically change the optical phase. The terahertz detector was fixed at a certain angle, and the power of the terahertz wave was measured. The power detected at an angle of $-20°$ (Figure 9, middle curve) and at an angle of $+10°$ (Figure 9, lower curve) both show periodical change of the power at 1 kHz and are upside down in relation to each other, which indicates that the terahertz beam is steered between $-20°$ and $+10°$ [16]. At above 1 kHz, the detected power decreased with the repetition frequency, which is consistent with the response speed of the thermo-optical phase shifter.

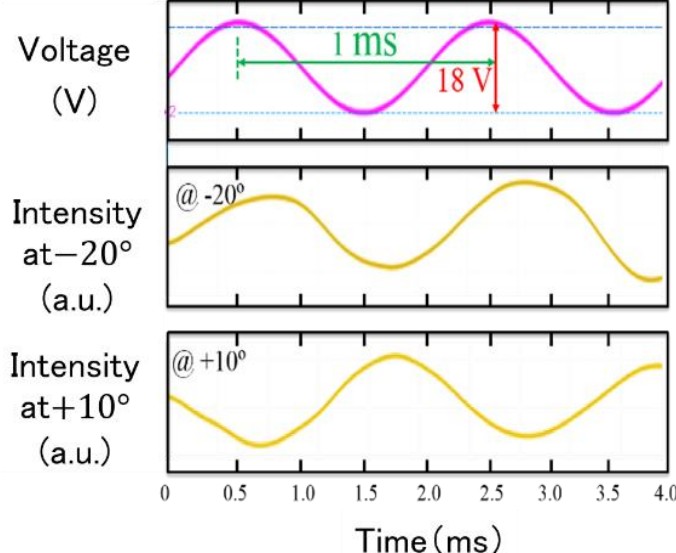

**Figure 9.** Applied voltage to the integrated phase shifter (**upper**), and intensity of 300-GHz wave at $-20°$ degree (middle) and $+10°$ degree (**lower**) during dynamic beam steering.

## 5. Application of Photonic Beam Control for High Security Wireless Transmission

The need of highly secured wireless communication is increasing, and thus, its encryption technology becomes one of the most important issues [17–19]. Beam forming and steering provide wireless communication with a secure physical layer and, in combination

with an upper layer algorithm, enhance overall communication security. For a more highly secured connection, we devised a wireless transmission system based on coherent detection between two RF data sequences utilizing beam control [20,21]. As shown in Figure 10, at the system, the data sequence to be sent (original data sequence) is divided into two data sequences (Data 1 and Data 2) by an encrypter so that the result of the AND between Data 1 and Data 2 coincides with the original data sequence. Two pairs of lightwaves (Lightwave pair 1 and Lightwave pair 2) are modulated by Data 1 and Data 2, respectively, and these are sent to the two transmitters via optical fiber. Here, the frequency difference between the two lightwaves of Lightwave pair 1 is not the same as that of Lightwave pair 2. At the transmitters, the terahertz beams (THz 1, THz 2) with the data sequences generated by the arrayed UTC-PDs are steered by optical phase control and they are overlapped with each other at a target position. A coherent receiver at the target position generates the heterodyned signal with a frequency difference between THz 1 and THz 2. Figure 11 shows the diagram of the processed data. Finally, an envelope detector reveals the overlapped data which results in the AND between Data 1 and Data 2 or the original data sequence. Deducing the original data requires Data 1 and Data 2 to be in the same data phase. In other words, the correct data sequence (original data sequence) can be deduced at a limited area within the order of the data bit length. Thus, the tolerance of this area is inversely proportional to the data rate; for example, it could be an order of centimeters for a 10-Gbit/s data rate. Therefore, the risk of interception can be greatly reduced using this system.

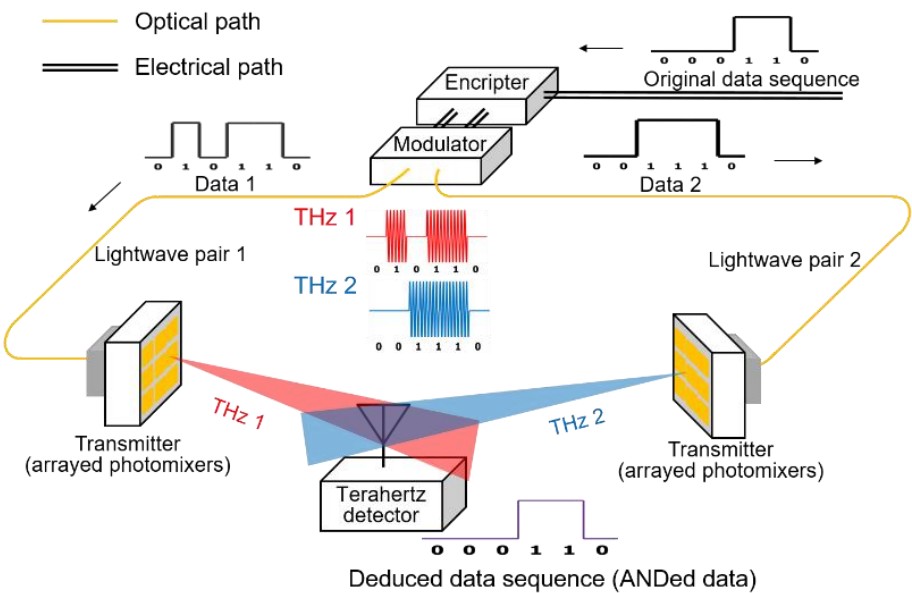

**Figure 10.** Secured wireless transmission system based on coherent detection between two RF data utilizing beam control.

For feasibility confirmation, the system was constructed as shown in Figure 12. Two lightwaves with a frequency difference of 300 GHz (Laser 1 and 2) were combined by optical coupler 1 (OC 1), and the other two lightwaves with a frequency difference of 310 GHz (Laser 3 and 4) were combined by optical coupler 2 (OC 2). An optical intensity modulator (IM) was used to modulate the intensity of each pair of lightwaves with the same 500-Mbit/s pseudo random pattern. The modulated lightwaves were divided by optical splitters (OSs) and inputted to two arrayed 0.5-mm-pitch UTC-PDs/antennas. One arrayed UTC-PD/antenna emitted a 300-GHz terahertz wave (THz 1), and the other emitted a 310-GHz terahertz wave (THz 2). They were transmitted 100 mm in space and were heterodyne-detected by a terahertz receiver. The beam angles of THz 1 and THz 2 were changed by the ODL inserted in each optical path so that THz 1 and THz 2 overlapped at the receiver. From the signal after heterodyne detection, the waveform was deduced by an envelope detector and observed by an oscilloscope (OSC). To confirm the AND operation

between THz 1 and THz 2, time delays of 7 ns and 10 ns were added by inserting optical fiber delay lines at the optical path of THz 1 [22].

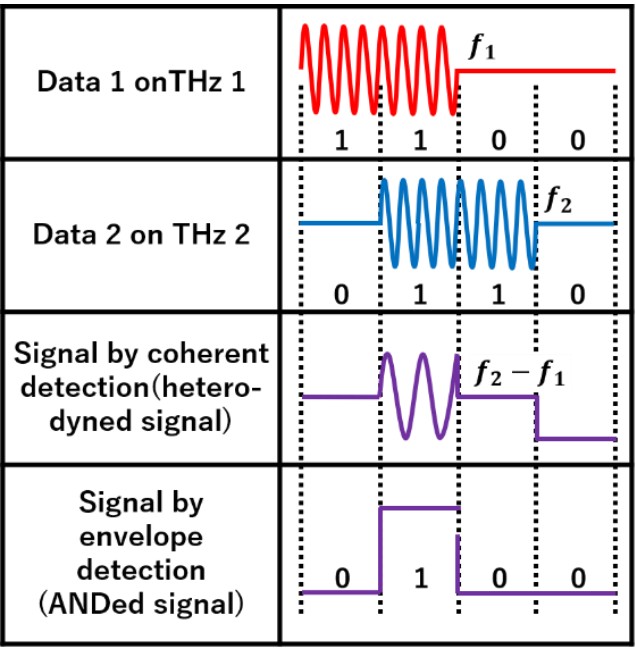

**Figure 11.** Secured wireless transmission system based on coherent detection between two RF data utilizing beam control.

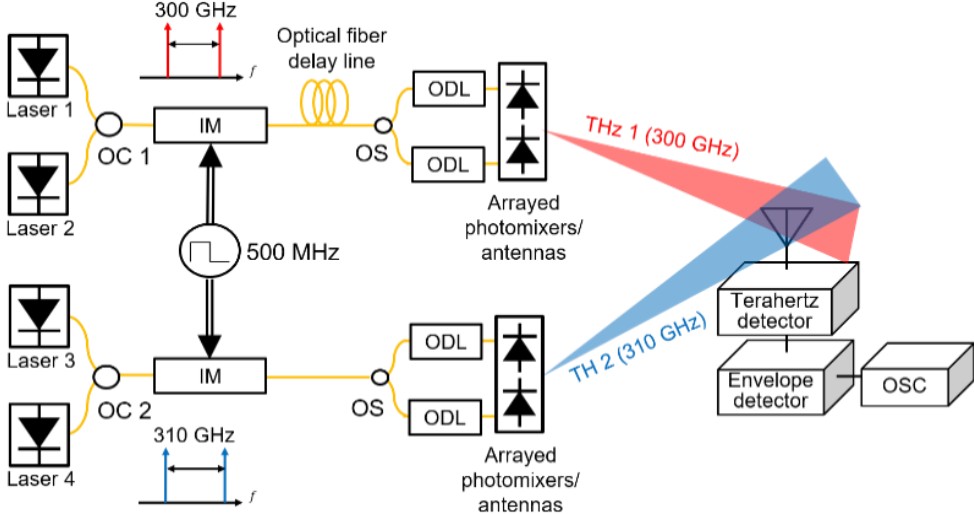

**Figure 12.** Experimental configuration for feasibility demonstration of the secured wireless communication system.

The observed waveforms of THz 1 and THz 2 are shown in the upper and middle graphs, and the ANDed signal is shown in the lower graphs in Figure 13. In the cases of both 7 ns and 10 ns, the "1" in the AND signal is generated only where "1" on THz 1 and THz 2 is overlapped. This feature is clearly confirmed by comparing the width of "1" at the areas marked "A" and "B".

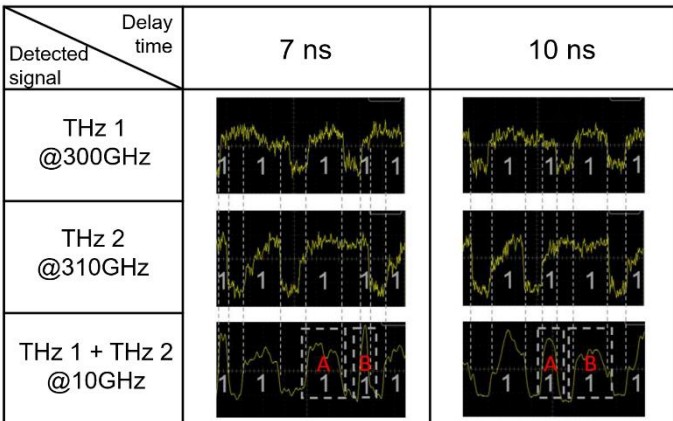

**Figure 13.** Observed waveforms of THz 1 (**upper**), THz 2 (**middle**) and THz 1 + THz2 (**lower**).

## 6. Discussion

The arrayed terahertz-wave generation based on photomixing showed a great impact on enhancing and concentrating the power because signals are easily copied, phase-tuned and distributed into the array in the optical domain. The experimental result showed that the peak power of the radiated terahertz wave is proportional to the square of the number of elements in the array. To tune the phase of the terahertz waves, either $-(k_2 x_2 - k_1 x_1)$ or $(\varphi_2 - \varphi_1)$ in (5) should be changed. For changing the former, the ODL was used in the experiment. The ODL changes an optical path length and thus, acts as a true time delay, which realizes terahertz-wave phase change independent of the frequency. On the other hand, for changing the latter, the optical phase shifter was used, which does not act as a true time delay or causes a frequency-dependent terahertz-wave angular distribution. We estimated that the peak angle of the beam is shifted by less than 3 degrees even with 20-GHz frequency change, which will not influence quality of 10- or 25-Gbit/s data.

Demonstration of the wireless transmission system based on coherent detection between two intensity-modulated RF data sequences showed the feasibility of AND operation between two RF signals. In combination with an upper layer encryption algorithm, it will enhance overall communication security. If phase modulation can be applied to this system, exclusive OR (XOR) operation can also be possible at the receiver, which would result in an even more secure system. For phase modulation, phase stabilization or digital signal processing is a key technology.

## 7. Conclusions

Photonics-assisted technologies for terahertz-wave generation, with the application of arrayed photomixers, are attractive because the phases of the terahertz waves can be tuned in the optical domain. We successfully combined and steered terahertz waves to make a beam at 300 GHz. In addition, using electrically controlled arrayed phase shifters on a silica-based planar lightwave circuit, we demonstrated the feasibility of high-speed beam steering. As an application for utilizing the advantages of terahertz wave beams, we proposed a secured wireless communication system, in which the receiver deduces ANDed data sequence from two terahertz beams and demonstrated the fundamental mechanism in a feasibility experiment.

**Funding:** This work was supported by the MIC/SCOPE #195010002, the Collaborative Research Based on Industrial De- mand/JST Grant Number JPMJSK1513, and JSPS KAKENHI Grant Numbers JP19K21977, JP19H02201, JP20H00253.

**Acknowledgments:** The author thank T. Nagatsuma, Osaka Univ. and H. Kanaya, Kyushu Univ. for continuous discussions.

**Conflicts of Interest:** The authors declare no conflict of interest.

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
