# Peer review of "Photonics-Assisted Terahertz-Wave Beam Steering and Its Application in Secured Wireless Communication"

_photonics, doi:10.3390/photonics9010009_

Round 1
Reviewer 1 Report
- The theoretical derivation of Beam Steering for Thz should be provided.
- The Thz should be compared with mmWave system, such as literatures in following: "Spatial- and Frequency-Wideband Effects in Millimeter-Wave Massive MIMO Systems" and "Low Complexity Interference Alignment for mmWave MIMO Channels in Three-Cell Mobile Network", etc.
- The author state that coherent receiver is adopted, is there any result of througput or BER?
- The presentation needs polishing.
Reviewer 2 Report
In this paper, the author proposes a method to generate terahertz waves by two lasers and a photomixer. Meanwhile, by tuning in the optical domain, the radiation pattern can be controlled. Moreover, a secure wireless transmission scheme is proposed and demonstrated.
However, there are several comments the authors shall clarify.
(1) In the experimental setup shown in Fig. 3, do the light signals from the two lasers propagate into the photomixers by optical fiber or free space channel? If fibers are used, then wave numbers in Eq.(4) may be incorrect.
(2) In the case of asymmetric path lengths, does there exist any influence on the scheme (i.e. Eq. (3)). A note should be given.
(3) In the second paragraph of section 4, it would be better that a schematic diagram is added.
(4) Can the proposed wireless transmission scheme prevent fake source attack?
Reviewer 3 Report
- The authors use two terms, i.e., "beam forming" and "beam steering", in the Introduction. It will be helpful to include their brief definitions and explain their differences before using them in the first paragraph of Introduction.
- Since the definition of coherent detection used in this work (coherent combining two incoming signals) is a bit different from conventional definitions in radio or optical communications, it will be helpful if the authors can briefly define such a unique coherent detection scheme and explain why could further improve on the security already in the Introduction.
- A few measurements shown in Fig. 5 have faster power roll-offs or dips. Please explain.
- It has been a concern to use phase shifters instead of true time delays in broadband communication systems (particularly for phase modulated broadband signals) for beam steering. Can the authors comment on, and if possible, quantify on this aspect?
- The results of the security wireless transmission is very interesting. But the current description seems to brief to fully understand the principle. Can the authors elaborate on why the combination of the THz 1 and THz2 binary data (0 and 1) result in 2 levels (0 and 1) in stead of 3 levels (0, 1 and 2)? How does the heterodyne receiver work to achieve this?
